# Quantifying Southern Hemisphere dust sources during the Last Glacial-Interglacial Transition using rare earth elements in the EPICA Dome C ice core

Sibylle Boxho<sup>1,2</sup>, Aubry Vanderstraeten<sup>1,3</sup>, Nadine Mattielli<sup>1</sup>, Goulven G. Laruelle<sup>2</sup>, Aloys Bory<sup>3</sup>, Paolo Gabrielli<sup>4</sup> and Steeve Bonneville<sup>2</sup>

- <sup>1</sup> Laboratoire G-Time, Département Géosciences, Environnement et Société (DGES), Université Libre de Bruxelles (ULB), Av. F. Roosevelt, 50 (CP 160/02), Brussels, 1050, Belgium.
- <sup>2</sup> Biogéochimie et Modélisation du Système Terre, Département Géosciences, Environnement et Société (DGES),
   Université Libre de Bruxelles (ULB), Brussels, 1050, Belgium.
  - <sup>3</sup> Laboratoire d'Océanologie et de Géosciences UMR 8187 LOG, Univ. Lille, CNRS, Univ. Littoral Côte d'Opale, IRD, F-59000 Lille, France.
  - <sup>4</sup> Italian Glaciological Committee, c/o University of Turin, Turin, Italy

Corresponding authors: Sibylle Boxho sibylle.boxho@ulb.be; Steeve Bonneville steeve.bonneville@ulb.be

- 15 **ABSTRACT.** Dust deposits in ice cores provides a valuable archive of past atmospheric circulation, offering insights into climate dynamics during key climate transitions such as the last glacial termination. Here, we present a novel continuous, high-resolution reconstruction of dust provenance in the EPICA Dome C (EDC) ice core from 33.7 to 2.9 kyr BP, based on Rare Earth Element (REE) patterns. Using a statistical unmixing algorithm on 241 samples, we quantify for the first-time contributions from key Southern Hemisphere dust sources. During the Marine isotope 3 (MIS3), the Last Glacial Maximum (LGM) and Heinrich Stadial 1 (HS1), dust at EDC was dominated by Patagonia sources (~65-75%), with secondary inputs from Australia, Southern Africa, New Zealand, and Puna-Altiplano. After ~14.5 kyr BP, the dust assemblage shifted toward greater inputs from low-latitude sources - Australia, Southern Africa, and the Puna-Altiplano - culminating during the Holocene with 25 more variable compositions and reduced Patagonian input (~49% on average). This transition is supported by Sr-Nd isotopic calculations and aligns with changes observed in other East Antarctic ice cores. Comparison with the EDML ice core reveals overall agreement in major sources and timing of the shifts but also highlights regional variations in the secondary contributions, with EDC showing more consistent inputs from Australia and EDML from Southern Africa. We attribute the shift in provenance 30 to long-lasting hydrological changes in Patagonian river systems and a rapid submersion of the
- Patagonian shelf around 14.5 kyr BP, linking dust composition to eustatic sea-level rise.







#### 1. Introduction

Antarctic ice cores are exceptional archives of past atmospheric conditions, preserving high-resolution records of dust deposition that provide critical insights into the evolution of global climate and atmospheric circulation (Lambert et al., 2008). Over recent decades, studies on these ice cores have revealed the connection between dust fluxes, ocean export productivity, and the development of deep glaciations, connections thought to be driven at least in part by dust-borne iron fertilization of phytoplankton in the High Nutrients Low Chlorophyll (HNLC) regions of the Southern Ocean (Martínez-García et al., 2014). Enhanced iron delivery by atmospheric dust during glacial periods are modelled to have lowered the atmospheric pCO<sub>2</sub> by ~20 to 40 ppmv, accounting for a significant portion of the glacial-interglacial pCO<sub>2</sub> difference (Lambert et al., 2015; Lambert et al., 2021). However, large uncertainties in those estimations remain due to limited knowledge of dust emissions from specific Potential Source Areas (PSAs) during the Last Glacial-Interglacial Transition (LGIT). Indeed, dust deposition rates are primarily influenced by both the geographic location of dust sources and the residence time of dust in the atmosphere (Lambert et al., 2008). This complexity is further compounded by inter-source variability in iron content and solubility, shaped by local weathering processes and the intensity of long-range atmospheric processing (Qu, 2016; Shi et al., 2009, Shi et al. 2011, Shi et al. 2015). Therefore, identifying and quantifying the contributions of individual PSA is essential to help better constrain models of dust-driven iron fertilization as well as improving paleoclimate reconstructions, and thus advancing our understanding of changes in Southern Hemisphere westerly winds, which play a crucial role in atmospheric and oceanic circulation (Engelbrecht et al., 2019).

Antarctic ice cores, especially those collected from high-altitude, inland drill sites, provide continuous and well-dated records of long-distance dust transport (Grousset and Biscaye, 2005; Basile et al., 1997; Delmonte et al., 2010; Gili et al., 2017). Geochemical characterization of sediments in regions up-wind from deposition sites is essential to the process of source identification, these PSAs serve as end-members against which to compare samples of dust from ice cores. The most prominent PSAs in the Southern Hemisphere are Southern South America (SSA), Southern Africa (SAF), Australia (AUS), New Zealand (NZ), and ice-free regions of Antarctica (ANT). While isotopic fingerprinting has been the preferred method for tracing dust sources for decades, its application is limited by the amount of dust (thus, ice volumes) required for analysis, which reduces temporal resolution, particularly in remote, high altitude site such as EDC where dust fluxes is low and *a fortiori* during interglacial periods where dust fluxes are up to 25 times lower than LGM fluxes (e.g. Delmonte et al., 2008; Delmonte et al., 2017).

Another issue lies in the difficulty to discern multiple source contributions, especially when the isotopic fields overlap. Moreover, the dust mixing during transport adds another layer of complexity in the interpretations of dust isotopic compositions in terms of provenance (Gaiero et al., 2004; Delmonte et al., 2020). All of the above-mentioned factors render the quantification of dust source contribution complex to achieve using isotopic fingerprinting.






Rare Earth Element (REE) patterns offer a promising complementary approach for dust provenance studies owing to the fact that the 14 concentrations/observations contain a lot more discriminative information than REE ratio. Moreover, REE analyses require smaller sample sizes, enabling higher temporal resolution and precision. Previous studies using REE patterns in Antarctic ice cores (e.g., Gabrielli et al., 2010; Wegner et al., 2012) achieved multi-decadal resolution but could not fully resolve dust provenance. Recently, using the REE dataset from Wegner et al. (2012), we presented the first quantitative apportionment algorithm based on REE pattern fitting, capable of disentangling contributions from multiple dust sources across the Southern Hemisphere (Vanderstraeten et al., 2023). Our results for EDML ice core located in the Atlantic sector of East Antarctica reveal that the dust composition remained relatively constant across LGM and HS1 periods, being predominantly composed of Patagonian dust, up until ~14.5 kyr BP when a sudden drop in Patagonian contribution (below 50% on average) was evidenced while the dust contributions from low-latitude PSA, particularly from Australia and Southern Africa, sharply increased (Vanderstraeten et al., 2023). Our dust provenance record in EDML also revealed that early Holocene dust characteristics (up until 7.5 kyr BP) were much more variable than LGM dust. Based on this new provenance record, we argued that the abrupt submersion of the Patagonian shelf by the sea-level rise at ~14.5 kyr BP and long-lasting hydrological rearrangement of Patagonian rivers following deglaciation, played an important role in reduction of Patagonian dust contribution and hence in the dust composition recorded in EDML, highlighting a potential link with eustatic sea level.

Here, we applied our 'DEPOT' model - to the REE pattern dataset measured - in the EPICA Dome C ice core (Gabrielli et al., 2010) to provide a high-resolution quantitative and continuous reconstruction of dust provenance in the Indian Ocean sector of the East Antarctica during the LGIT from 33.6 kyr to 2.8 kyr BP. Our study aims at (1) establishing a quantitative record of dust provenance in EDC throughout the LGIT and particularly during the Holocene for which there is no consensus (e.g. Paleari et al., 2019; Coppo et al., 2022; Delmonte et al., 2007), (2) confronting and validating the EDC results from our REE pattern-based model against Sr-Nd isotopic data published in the literature, (3) examining shifts in dust provenance at EDC in relation to regional climatic changes affecting the PSA, and (4) contrasting this new EDC provenance with the coeval record in EDML in the Atlantic sector of the Antarctic continent in order to assess spatial variability of dust deposition across East Antarctica during the LGIT.

### 2. Materials and methods

Dust rare Earth element Pattern Over Time ("DEPOT" algorithm)

To identify and quantify dust source contributions to ice core camples, we developed a numerical approach based on a constrained least-squares algorithm. This method solves a system of linear equations using only positive coefficients, as source contributions correspond to physical proportions of






dust and cannot take negative values. The fit between the REE patterns measured in the EDC ice core and literature reported patterns from known potential source areas (PSA) across the Southern Hemisphere is then optimized using the coefficients to quantify the mixing between all PSAs. A detailed description of the methodology is available in Vanderstraeten et al. (2023). Briefly, our analysis used the REE dataset from the EDC ice core, originally published by Gabrielli et al. (2010) comprising 294 Dust samples from Ice Cores (DIC). The dataset spans from the Late Holocene to the end of MIS 3 (2852 to 33699 yr BP), with an average temporal resolution of approximately 100 years, ranging from 60 to 140 years between adjacent samples. The age model follows the EDC3 chronology (Parrenin et al., 2007). Each sample integrates between 2–3 years of deposition during the Holocene and 4–5 years during the LGM. A sampling gap exists between 21.7 and 27.3 kyr BP, where only four samples were available.

Out of the original 294 REE patterns, 15 were excluded due to missing data for one or more of the 14 REEs analyzed. Additionally, gadolinium (Gd) was systematically removed from all samples due to its low and often unreliable concentrations adding more noise than discriminatory power to the calculations. The remaining 279 REE patterns (each comprising 13 REE concentration values) were normalized to the Upper Continental Crust (UCC) composition (Rudnick and Gao, 2003) and corrected using element-specific factors (cf<sub>i</sub>) as described in Gabrielli et al. (2010) and Vanderstraeten et al. (2023) (Eq. 1). These element-specific correction factors adjust for partial dissolution (with ultrapure HNO<sub>3</sub>) resulting from the acid-leaching extraction dissolution method used during REE quantification. Since this method does not fully dissolve all mineral particles, we applied cf<sub>i</sub> derived from parallel measurements of fully digested EDC samples to obtain total REE concentrations allowing fitting with REE patterns from PSA (Gabrielli et al., 2006; Vanderstraeten et al., 2023).

$$125 cf_i = \frac{\left[REE_{full\ digestion}\right]_i}{\left[REE_{acid-leaching}\right]_i} (1)$$

To fit the REE patterns from EDC, we compiled a comprehensive database of 217 REE patterns from well-established dust source regions across the Southern Hemisphere (Figure S1, and Table S1). This dataset (Table S2) includes samples from Southern South America (SSA), subdivised into Patagonia (PAT; 39 dust sources, DS), Central Western Argentina (CWA; 5 DS), and the Puna-Altiplano Plateau (PAP; 14 DS from North and South Puna and Southern Altiplano), as well as from Southern Africa (SAF; 23 DS from Namibia and Botswana), Australia (AUS; 106 DS, predominantly from the central and southeastern regions), New Zealand (NZ; 25 DS, from the South Island), and Antarctica (ANT; 5 DS from the McMurdo Dry Valleys, representing ice-free sources). As with the EDC ice core samples, all PSA REE patterns were normalized to the UCC composition of Rudnick and Gao (2003) and Gd values were removed. Dust source apportionment was conducted using a constrained least-squares approach, wherein the REE pattern of each EDC sample was modeled as a linear combination of PSA end-members. The model estimates the respective Contributions of each Dust Source (CDS1...217) by solving the system of matrixes (Eq.2) with non-negativity constraints, minimizing the resulting values,






and ensuring that all contributions are physically meaningful (i.e., no negative values). This was implemented using the Isquonneg function in MATLAB.

$$\begin{bmatrix}
DS_{La}^{1} & \cdots & DS_{La}^{217} \\
\vdots & \ddots & \vdots \\
DS_{Lu}^{1} & \cdots & DS_{Lu}^{217}
\end{bmatrix} X \begin{pmatrix} CDS^{1} \\
\vdots \\
CDS^{217}
\end{pmatrix} - \begin{pmatrix} DIC_{La} \\
\vdots \\
DIC_{Lu}
\end{pmatrix}_{2}^{2}$$
(2)

In practical terms, the source apportionment model involves solving a system where the input matrix-comprising "n" rows (one for each UCC-normalized REE) and 217 columns (one for each <u>dust source</u> sample, DS)- is multiplied by a vector of 217 components (CDS<sup>1...217</sup>). In Vanderstraeten et al. (2023), n was equal to 14 but is only 13 in the present study because of the exclusion of Gd. For any given time step. The vector of 217 components represents the proportional contribution of each DS required to best approximate the REE signature of a given <u>dust ice core</u> (DIC<sub>la-lu</sub>) sample. The solution to this system is the set of CDS values that minimizes the residual between the modeled REE composition (i.e., the product of the input matrix and the CDS vector) and the measured DIC REE pattern. Importantly, only a subset of the 217 dust sources typically contributes to the best-fit solution for a given DIC sample—that is, only sources with CDS > 0. Sources with CDS equal to 0 are excluded from the fit, indicating no detectable contribution to that specific DIC sample.

To evaluate the impact of uncertainties in REE concentrations in DIC samples and their associated correction factors (cf<sub>i</sub>) on our model outputs, we performed Monte Carlo (MC) simulations for each DIC sample. Analytical uncertainties associated with REE concentrations in EDC dust range between 2% and 10% (1 relative standard deviation), depending on the specific REE - *Dust<sub>i</sub>* (Gabrielli et al., 2006). In each MC simulation, cf<sub>i</sub> and *Dust<sub>i</sub>* values were randomly selected following normal distributions constrained by their respective mean values and variability to represent these uncertainties. The algorithm then recalculated the best-fit source contributions (CDS values) by minimizing Eq. (1). Each DIC sample was subjected to 2000 simulations and the ensuing CDS results (in % of total deposition) generate a probability density function (PDF) for every DS which were subsequently aggregated by PSA or specific sub-regions within PSA. The choice of 2000 MC simulations is a tradeoff between computation cost and representativity of the uncertainty propagation in the PDFs but the use of a larger number of simulations (5000) does not significantly affect the results. The median of each PSA-specific PDF was taken as the best estimate of its contribution to the DIC sample's total dust load. Additionally, we used the total dust influx to EDC—derived from non-sea-salt calcium flux (nssCa<sup>2+</sup> in μg m<sup>-2</sup> yr<sup>-1</sup>; from Fischer et al., 2007) — to convert PSA contributions into absolute dust fluxes (Figure S2).

To assess the quality of each fit, we computed the correlation coefficient (R) between the DIC and modeled REE patterns for every MC simulation. The 2000 R values per DIC sample formed a PDF from which the median R value was taken as the representative measure of fit quality. We determined that R  $\geq 0.67$  corresponds to a statistically significant match (p 







#### 3. Results

Out of the 279 DIC samples treated by our algorithm, 38 did not meet the R threshold (R≥0.67) and were excluded from the provenance record. These excluded samples are all younger than 14 kyr BP and have very low REE concentrations near the detection limit. Such low concentrations are consistent with minimal dust input to EDC after HS1 (Gabrielli et al., 2010; Fischer et al., 2007). At these levels, element-specific analytical noise exacerbated by varying ionization efficiencies and UCC normalization – distorts the REE patterns, rendering fitting meaningless. For the remaining 241 DIC samples (86%), the PSA mixes calculated by the model were considered meaningful compared to the uncertainties associated with the measurement and characterized by an average correlation coefficient of R = 0.852. These DIC samples span from 33.7 kyr BP to 2.9 kyr BP and form the basis of the dust provenance record presented here. Figure 1 and Table S1 summarize the relative contributions (in % of total deposition and flux) from each PSA across major climatic intervals: the end of Marine Isotope Stage 3 (MIS3: 57–29 kyr BP), the Last Glacial Maximum (LGM: 29–18 kyr BP), Heinrich Stadial 1 (HS1: 18–14.7 kyr BP), the Antarctic Cold Reversal (ACR: 14.7–12.9 kyr BP), the Younger Dryas (YD: 12.9–11.7 kyr BP), and the Holocene (11.7–2.8 kyr BP; with Early Holocene 11.7-7.5 kyr BP and Late Holocene 7.5-2.8 kyr BP) (Veres et al., 2013).

PAT is the dominant dust source throughout most of the record. It contributes for 65-70% (average contributions) of the total dust deposition during the end of MIS 3 and the LGM, increasing to 75% during HS1 (Table S5). Overall, the PSA dust assemblage during HSI is marked by a stable and homogeneous apportionment. During the ACR and YD, contributions of PAT declined to 69% and 63% (averages contributions), respectively, with the downward trend continuing into the early Holocene. Three distinct minima in PAT contributions occur at 10, 8.6, and 6.3 kyr BP. Despite increased variability during the Holocene, PAT remains the primary source, contributing to 47-51% of the total mix on average. Among the 39 DS from PAT, our algorithm highlights three specific REE patterns collected in or near the Valdes Peninsula. The samples represent aeolian materials gathered during dust sampling campaigns and therefore reflects regional mixed input rather than single lithological sources. Their mafic-like compositions, reflected by LREE (low rare earth elements) depletions, show positive Eu anomalies relative to UCC (Gili et al., 2017; Gaiero et al., 2004).

Overall, other Southern Hemisphere PSAs contribute to EDC dust deposition with sources from: AUS, SAF, PAP and NZ, in decreasing order of importance. During MIS3, LGM and HS1, AUS accounted for 9, 10 and 11% of total dust deposition on average, respectively. Its average contributions rose after 14.7 kyr BP, reaching 11% during the ACR, ~15% during the YD, and reaching 23% during the Holocene, with two prominent maxima centered around 9.5–10 kyr BP and ~6.3 kyr BP (Figure 1). SAF maintained a relatively stable contribution ranging between ~5 and 9% on average during MIS3, LGM, HS1 and ACR, followed by a sharp increase during the YD with 15%. Although SAF input drops to minimal




levels (~2%) between 9.5 and 12 kyr BP, it rebounds in the subsequent Holocene, averaging ~11%. NZ was the second-largest contributor during the LGM with an average of 14% but its influence steadily declines over time: 7% during HS1, and only 6.5%, 3%, and 6% during the ACR, YD, and Holocene, respectively. The PAP contributes modestly during the MIS3, LGM and HS1, with average values below 4%, increasing to ~5% during the ACR and YD, primarily from the South Puna region. In the Holocene, a compositional shift occurs, marked by the emergence of South Altiplano dust, which even becomes dominant in several samples, contributing over 90% at 11.2, 10, 8.6, 6, and 3.5 kyr BP. In contrast, contributions of South Puna remained consistently below 20% during this interval (Figure 1). On average, PAP accounts for 11% of the total Holocene dust input to EDC ice core (Table S5). Other PSA, including CWA and ANT—only contributed negligibly (data not shown on Figure 1, see Table S1, Table S5).

#### 4. Discussion

In this study, we present the first continuous and quantitative record of dust provenance in the Epica Dome C ice core spanning the entire LGIT (Figure 1). Our results indicate that, during the LGM, atmospheric dust deposition at EDC was dominated by inputs from PAT, with smaller but measurable contributions from NZ, AUS, SAF and PAP dust (in decreasing order of importance). This composition remained relatively stable throughout HS1, despite a pronounced decline in total dust deposition after ~18 kyr BP. A shift in dust source contributions becomes evident after ~14.5 kyr BP, characterized by larger proportions from AUS, PAP, and SAF, while the relative contributions from PAT and NZ declined. In the following sections, we (1) evaluate the consistency of our provenance reconstruction with the isotopic composition of dust measured in EDC and other East Antarctic ice cores; (2) discuss the shift in dust provenance occurring along the LGIT in conjunction with regional climatic evolution (3) compare our continuous provenance record from EDC with that of the EDML ice core (Vanderstraeten et al., 2023).

## 4.1 Isotopic and REE compositions of dust in EDC

Our calculations highlight a major shift in dust provenance occurred at the onset of the ACR, marking a transition from a dust assemblage dominated by high-latitude sources—namely PAT and NZ—to a more diverse mixture with significantly increased contributions from low-latitude sources, including AUS, SAF and PAP. This shift already describes by Gili et al. (2022) and Vanderstraeten et al. (2023), appears to be consistent with the changes in isotopic composition of dust deposited at EDC between glacial and interglacial periods. For instance, Delmonte et al. (2004) reported  $^{87}$ Sr/ $^{86}$ Sr ratios of  $\sim$ 0.708 and  $\epsilon$ (Nd) values ranging from -1.66 to -2.46 for LGM dust at EDC, whereas Holocene dust typically displays more radiogenic  $^{87}$ Sr/ $^{86}$ Sr ratios ( $\sim$ 0.710) and more negative  $\epsilon$ (Nd) values (-4.51 to -5.68). Similar shifts in dust isotopic composition observed in dust from the Vostok ice core further support the hypothesis of a change in source regions (Basile et al. 1997; Delmonte et al. 2007). To compare our REE-based source






apportionment with isotopic evidence, we calculated the expected bulk <sup>87</sup>Sr/<sup>86</sup>Sr and ε(Nd) signatures of EDC dust from the modeled source contributions (Figure 2). For each time interval of the LGIT, the relative contribution of each PSA (CONTR) was multiplied by its average Sr and Nd concentrations, [Sr]<sub>PSA</sub>, [Nd]<sub>PSA</sub>, to obtain concentration-weighted contributions, [X]<sub>mix</sub> (Table S3, *e.i.* Eq.3).

$$[X]_{mix} = \sum (CONTR_{PSA} * [X]_{PSA})$$
(3)

Where "X" is the Sr or the Nd concentration. The composite isotopic ratios (CIR<sub>mix</sub>) were then calculated for ( $^{87}$ Sr/ $^{86}$ Sr) and  $\epsilon$ (Nd) in Eq.4, where the mean isotopic signatures is represents by CIR  $_{PSA}$ .

250 
$$\operatorname{CIR}_{\operatorname{mix}} = \sum (\operatorname{CIR}_{\operatorname{PSA}} * \operatorname{CONTR}_{\operatorname{PSA}} * \frac{[X]_{\operatorname{PSA}}}{[X]_{\operatorname{mix}}})$$
 (4)

The modeled isotopic signatures for the end of MIS3/LGM, HS1, ACR, and the YD (represented on Figure 2 by blue, cyan, green, and yellow filled symbols) cluster around lower <sup>87</sup>Sr/<sup>86</sup>Sr and higher ɛ(Nd) values, consistent with a provenance dominated by less weathered, mafic lithologies in accordance with large Patagonian and New Zealand contributions. In contrast, the modeled and measured Holocene isotopic compositions (red filled and open symbols, respectively) both show a broader range, trending toward more radiogenic Sr and less radiogenic Nd isotopic values, indicative of a more heterogeneous and weathered source assemblage—consistent with greater inputs from AUS, SAF, and PAP (Figure 2). Overall, the close agreement between our modeled isotopic compositions and those measured in EDC for both glacial and interglacial intervals supports the robustness of our DEPOT algorithm and reinforces the interpretation that a distinctive shift in dust provenance occurred over the LGIT.

# 4.2 Persistance of high-latitude dust source to EDC before 14.5 kyr BP

During cold periods, glacial outwash plains played an important role in sediment mobilization and dust supply to the atmosphere in the Southern Hemisphere (Bullard et al., 2016). Extensive ice sheets produced large amounts of fine sediment which were effectively spread by wide, migrating braided river systems. Exposed to arid, cold conditions and strong westerly winds, those deposits would routinely dry out and provide a major source of dust for long range transport (Sugden et al., 2009). Such conditions were especially prevalent in PAT where the Patagonian Ice Sheet reached its maximum spatial extent between ~34 and 29.4 kyr BP, covering much of the Andes with an ice mass 2090 km long and 350 km wide (Davies et al., 2020). Concomitantly, sea level was ~135 m lower than present-day, exposing additional land and expanding Patagonian outwash plains by ~763.10<sup>3</sup> km<sup>2</sup> relative to modern conditions (Vanderstraeten et al., 2023). These factors collectively explain the predominance and relative stability of PAT dust contributions and, by extension, those from SSA- to EDC during the LGM and MIS3. Isotopic studies consistently demonstrate the significance of Patagonian sources in glacial dust deposited across East Antarctica (EA) (e.g., Grousset et al., 1992; Basile et al., 1997; Gaiero, 2007; Gili et al., 2017; Delmonte et al., 2020). Model simulations also support this view, estimating that SSA and AUS






sources together supplied 70-90% (here, PAT+AUS is ~77%) of the dust reaching EA during the LGM (i.e., Albani et al., 2012; Krätschmer et al., 2022). Based on Sr–Nd isotopes, Coppo et al. (2022) estimated LGM dust composition over EA as 64% from SSA, 15% from Antarctica, 11% from AUS, 5% from SAF, and 5% from NZ.

Our results for SSA, which include PAT + PAP + CWA, closely match Coppo et al. (2022), with a combined contribution of ~70%, and ~10% from AUS (Figure 3; Table S5). However, we find a higher average NZ contribution (~11%), making it the second-largest dust source to EDC during the LGM. This contrasts with Koffman et al. (2021), who argued against significant NZ dust input to EA. Nevertheless, the extensive NZ South Island ice sheet, enlarged continental shelf (~62,000 km², see Vanderstraeten et al., 2023), and cold, windy LGM conditions likely favored enhanced NZ dust emissions capable of reaching Antarctica (Williams et al., 2015). Present-day forward air-mass trajectories indicate NZ dust could contribute 14-32 % to inland Antarctic deposition (Neff and Bertler, 2015) and our previous EDML ice core work detected NZ dust at ~6 % on average over the entire LGM. In contrast, PAP contributions during the LGM were minimal (~4 %), likely due to persistent wet conditions in the South Altiplano linked to the Sajsi and Tauca lake phases (Blard et al., 2011; Placzek et al., 2006). Consistently, Pampean Loess accumulation rates—largely sourced from PAP dust—were low during both the LGM and HS1 (Coppo et al., 2022). Overall, the LGM dust assemblage at EDC was dominated by high-latitude, fluvio-glaciogenic sources—PAT and NZ—which together accounted for ~79 % of total deposition. Lower-latitude PSAs (AUS, SAF, PAP) supplied the remaining ~21 % (Figure 3; Table S5). This dominance persisted into HS1, with high-latitude sources totaling 84 % despite a sharp decline in dust flux to EDC starting at ~18 kyr BP (Figure S2). The latter likely reflects intensified "rainout" during long-range transport, particularly in the mid-latitudes, acting as a barrier to polar dust delivery (Markle et al., 2018). This reduced transport efficiency coincides with rising temperatures between 18 and 15 kyr BP, as indicated by elevated  $\delta^{18}$ O values in EDC (Figure S2). The persistence of high-latitude dominance before 14.5 kyr BP reflect a transport regime sustained by extensive ice sheets, widespread outwash plains, and strong westerlies -a system that maintained a remarkably stable provenance pattern until the onset of the LGIT. The following section examines how this balance shifted under the warmer, wetter, and more variable climate of the deglaciation and Holocene (from 14.5 kyr BP until 2.7 kyr BP).

4.3 Holocene Variability in Dust Provenance and Importance of Regional Source Dynamics

During YD and throughout the Holocene, our unmixing model reveals a significantly larger variability in the dust assemblages reaching EDC as shown in Figure 1. This increased variability, seemingly higher frequency changes in sources, likely reflects several interrelated processes: (1) a decline in fluvioglaciogenic dust from PAT (and NZ) that had previously strongly dominated atmospheric deposition in EA prior to 14.5 kyr BP (see above for discussion in section 4.2); (2) a reduced transport efficiency







associated with enhanced «wet scavenging» due to rising temperatures; and (3) the lengthened transport trajectories inferred by increased deuterium-excess values, especially after 10 kyr BP (Figure S2), which suggest a northward shift in moisture sources from the mid-latitudes of Indian Ocean (Stenni et al., 2010).

Despite the overall increase in the variability of the dust composition after ~14.7 kyr BP, our results reveal a clear shift toward increased contributions from low-latitude sources namely AUS, PAP and SAF with average Holocene contributions of 23%, 11% and 11 % respectively (compared to 10%, 4% and 6% during LGM, Table S5). Conversely, average contributions from high-latitude PSA – PAT and NZ – decline during the Holocene to 49% and 6%, respectively. The DustCOMM model for present-day dust deposition in Antarctica estimates contributions from Southern South America (SSA) at 70 ± 20% (Kok et al., 2021) —compared to 61% in our record when combining PAT, CWA, and PAP. DustCOMM also predicts dust contributions from AUS and SAF at 18% (ranging from 6% to 36%) and 10% (ranging from 3% to 13%), respectively (Kok et al., 2021), which aligns reasonably well with our results for AUS and SAF. A similar transition from high- to low-latitude dust sources was observed by Coppo et al. (2022), although their model indicates a much more pronounced reduction in SSA input -down to 33%compared to our SSA estimate of 60% (Table S5). Their results show comparable contributions from AUS and SAF - i.e., 23% and 14%, respectively vs. 23% and 11%, here - and a larger Antarctic component (24%), specifically from the McMurdo Dry Valleys and nearby Southern Victoria Land volcanoes. Although our Dust Source (DS) database includes several McMurdo region sites (Section 4.2), our results do not indicate significant Antarctic contributions to EDC. This is consistent with the stable atmospheric conditions and prevailing katabatic winds over the East Antarctic Plateau, which hinder dust uplift and transport from lower-elevation McMurdo valleys to Dome C—a route involving >1,200 km of horizontal travel and 3,200 m of ascent. Furthermore, dominant winds in the McMurdo region blow from the east-southeast, opposite to the direction toward Dome C (Speirs et al., 2001).

The contribution of Patagonian dust to EDC showed a continuous decline after 14.5 kyr BP during the ACR, YD reaching minimum values at ~10 kyr BP (discussed below). Interestingly, in the mid Holocene, PAT contributions fluctuated widely between 10 and 6 kyr BP with peak contributions centered ~9 kyrs BP, between 7-8 kyr BP and after 6 kyr BP (Figure 1). Overall, our Holocene provenance record for PAT contributions is matching well with the SWW intensity proxies in Potrok Aike bog (see Figure 5). The 11 to 8 kyr BP period corresponds to a weakening of South Westerlies Winds (SWW) across the Southern Hemisphere (Fletcher and Moreno, 2012; Moreno et al. (2010); Lisé-Pronovost et al., 2015), possibly associated with a poleward shift of SWW in Patagonia (Quake and Kaplan, 2017). At first glance, this seems unfavorable for PAT dust emissions, however, this interval corresponds as well with a drier climate in eastern Patagonia evidenced by a widespread increase in paleofire occurrence in SSA (south of 30°S – Moreno et al., 2010; Power et al., 2008; Markgraf et al., 2007). Fires co-emit mineral soil-dust particle (Wagner et al., 2021), but also consume the soil-protecting








vegetation and induce the breakdown of larger soil aggregates into finer particles favoring aeolian deflation (Dukes et al., 2018). After 8.1 kyr BP, stronger SWW were recorded in S-E Patagonia (Potrok Aike bog 52°S; Lago Cardiel 48.9°S - Lisé-Pronovost et al., 2015; Quade and Kaplan, 2017), probably explaining the elevated PAT contribution in the 7-8 kyr BP interval. Apart from a well-marked drop in PAT contributions between 6-6.5 kyr BP (also present in Potrok Aike wind intensity record), strong SWW since 5 kyr BP recorded in South Patagonia correspond to 50-55% PAT contribution.

During the Holocene, the other important PSA from SSA i.e., Puna-Altiplano Plateau had its prime time in terms of contribution to EDC. Dust emission from PAP are primarily linked to aridification events and the desiccation of riverbeds, alluvial fans, and paleolake systems. As illustrated in Figure S3, South Puna (i.e., North Puna does not contribute) consistently provides dust throughout the entire record, including the Holocene, during which its contribution shows an increasing trend. In contrast, the Southern Altiplano begins contributing dust only after approximately 11.2 kyr BP, a timing that coincides with the final desiccation of paleolake Coipasa (Blard et al., 2011; Placzek et al., 2006) and the onset of persistent arid conditions in the region (Condom et al., 2004). The combined PAP dust assemblage—comprising a mafic component from the South Puna and a crustal component from the Southern Altiplano—is consistent with Sr, Nd and Pb isotopic compositions of Pampean loess reported by Coppo et al. (2022) that revealed a clear linear mixing relationship between these two endmembers. Overall, the relatively important contribution of PAP to EDC during interglacial periods is consistent with the previous Pb isotopic study (Gili et al., 2016).

As far as Australian dust contribution, our model identifies the Darling sub-basin as the dominant source region during the Holocene which was also suggested by De Deccker et al. (2010) using Pb isotopes. Hydrological reconstructions further support our results, showing that central Australia—specifically Lake Eyre and Lake Frome—experienced prolonged wet conditions between 14.5–4 kyr BP (Magee et al., 2004; Singh and Luly, 1991). These humid intervals would have lowered dust emissions from central Australia. In contrast, the Darling Basin underwent a significant dry phase from ~11.5 to 9.5 kyr BP, which coincides with a peak in AUS dust contributions to EDC. In addition, two consecutive humid phases (~13.5–11.5 and 9.5–7.5 kyr BP, potentially extending to 5.5 kyr BP) correspond to reduced dust fluxes (Gingele et al. 2007; Stanley and De Deckker, 2002). The mid- to late Holocene (post-7.5 kry BP) in AUS is characterized by increasing climatic instability with alternating wet and dry periods likely influenced by the El-Nino-Southern Oscillation (ENSO) or Interdecadal Pacific Oscillation (IPO) variability (Gingele et al., 2007; Stanley and De Deckker, 2002; Bullard and McTainsh, 2003; Lamb et al., 2009), which is consistent with large fluctuations in AUS contributions to EDC.

Southern African (SAF) dust sources are primarily associated with three major systems: the Makgadikgadi Complex (including Sua Pan), the Etosha Pan, and ephemeral river valleys along the Namibian coast (Wiggs et al., 2022; Gili et al., 2022; Prospero et al., 2002). During the Holocene, our








results show an increasing trend in SAF dust contribution to EDC ice core (Figure S4), which is consistent with an enhanced aeolian deposition of fine-grained material recorded in sediment from the Walvis Ridge in Southeast Atlantic Ocean suggesting strengthened dust transport from southern Africa suggesting strengthened dust transport from southern Africa (Stuut et al., 2002). Among SAF sources, the Namibian coast and ephemeral riverbeds were detected most frequently, though their overall contributions were smaller than those from the Etosha and Makgadikgadi systems. This pattern reflects the large spatial extent and the climatic sensitivity of ephemeral Namibian rivers, which experienced intermittent flow and channel aggradation in response to Holocene climate variability (Stone et al., 2010).

Notably, ephemeral riverbeds represent the majority of present-day dust plumes (Vickery et al., 2013). The Makgadikgadi Complex was a relatively frequent contributor with large but episodic inputs to EDC occurring between ~8.2 and 2.8 kyr BP. This timing coincides with a prolonged arid phase following the last highstand at ~17.1 ± 1.6 kyr BP, potentially interrupted by a brief humid interval around 8.5 ± 0.2 kyr BP (Brook et al., 2007; Franchi et al., 2022; Burrough et al., 2009). Dust contributions from the Etosha Pan were more sporadic but include few substantial inputs between ~12.2 and ~5.1 kyr BP, coinciding roughly with dry periods of the lake during the LGIT (~15.4–13.6, ~11.5–8.4, ~5.5–4.0 kyr BP (Brook et al., 2011). The intermittent signals from both Makgadikgadi and Etosha systems in EDC likely reflect their low-frequency, high-magnitude nature of their dust emissions (Wiggs et al., 2022; Bryant, 2003; Haustein et al., 2015; Vickery et al., 2013) potentially compounded by reduced atmospheric transport efficiency during the Holocene.

4.4 Spatial and temporal comparisons of EDC and EDML dust provenance records

The only continuous and quantitative dust provenance record in EA available for direct comparison with our EDC data comes from our recent study on the EDML ice core (Vanderstraeten et al., 2023). In Figures 4 and S5 (for millennial trends), we present a comparison of the relative contributions of PAT, NZ, SAF, AUS, and PAP from both cores over their overlapping interval shared by both datasets, i.e., between 7.5 and 27 kyr BP. It is important however to note that results for ACR and YD are tentative due to the limited number samples covering those periods in EDC (8 and 7 samples in ACR and YD, respectively, compared to 47 and 27 in EDML). At first glance in Figure S5, the dust provenance compositions of the two cores appear broadly similar, supporting the idea that atmospheric dust deposition across East Antarctica during the LGIT was relatively uniform. This hypothesis had been previously proposed for the LGM, MIS 4, and MIS 6 by Fischer et al. (2007) and Marino et al. (2009), based on similar geochemical dust emission patterns and major element signatures in both cores, suggesting a shared glaciogenic dust source from southern South America (SSA). A more detailed comparison, however, reveals subtle differences between EDC and EDML provenance contributions highlighting how atmospheric circulation modulates dust delivery to Antarctica (Delmonte et al., 2004).








Average dust source contributions during the LGM indicate that EDC received a greater proportion of NZ dust than EDML (+8%, i.e., [PSA average LGM contributions in EDC] - [PSA average LGM contributions in EDML] Figure 4), compensated by an equivalent reduction in contributions from PAT. During HS1, provenance compositions at both cores converge, with differences in PSA contributions not exceeding ~5%. In contrast, the ACR and YD intervals exhibit a slight enrichment of Patagonian dust (+4 and +6.5%) in EDC, primarily at the expense of Southern African and Australian contributions. In the Holocene, the discrepancies become more pronounced: EDC appears to receive substantially less PAT dust (-14%) but higher contributions of Australian dust (+12%) than EDML. These differences are consistent with the geographic positioning of the two sites: EDML is exposed to the storm track in the Atlantic sector, and thus well positioned to receive dust from Patagonia dust brought in the midtroposphere by the SWW. In contrast, EDC, located at the interior of EAP and the Indian Ocean sector, is comparatively insulated from such mid-latitude air masses incursions by the polar vortex. Nevertheless, planetary Rossby waves can disturb this polar vortex, allowing intrusions of mid-latitude or even low-latitude air masses. In such cases, dust from low-latitude PSA sources—either uplifted to high altitudes (~6–10 km a.s.l.) or directly deflated from elevated terrains such as the Puna–Altiplano can be entrained southwestward by the subtropical jet (STJ) into the polar front jet (PFJ) and transported to central Antarctica. For example, dust from the high-altitude Puna-Altiplano has been observed to follow this pathway beyond 50°S (Gili et al., 2017; Gaiero et al., 2013), while similar mechanisms over southeast Australia can inject dust into the STJ-PFJ system (Yang et al., 2024). Nguyen et al. (2019) demonstrated that prevailing southwesterly winds in central and southeastern Australia, combined with the Great Dividing Range, can uplift dust to altitudes of 4–6 km, enabling its long-range transport to the Indian Ocean sector of Antarctica. Other studies have similarly identified regions in Australia as significant contributors to both western and eastern Antarctica (De Deccker et al., 2019; Revel-Rolland et al., 2006; Vallelonga et al., 2002). The enhanced contributions of low-latitude sources to EDC during the Holocene reflect both warmer climate circulation patterns—favoring poleward STJ trajectories and interactions with PFJ (Ding et al., 2011) - and increased aridity in low-latitude regions (see above), which likely amplified dust emissions and transport from these sources.

In addition to the regional differences in dust provenance between EDC and EDML, both records reveal synchronous millennial-scale changes in dust source contributions. Notably, between 15 and 14 kyr BP, both ice cores show a marked decline in Patagonian dust contributions (Figure S5), accompanied by increasing inputs from Australia, Southern Africa and PAP. This shift from high- to lower-latitude dust sources is effectively decoupled from the earlier, more gradual yet massive decline in total dust deposition that began around 19-18 kyr BP and continued until ~15 kyr BP in both cores (Figures S2 and S5). As detailed in Vanderstraeten et al. (2023), this provenance shift coincides with several major large-scale events: the large Antarctic Iceberg Discharge AID6, the onset of the Bølling interstadial in the Northern Hemisphere, and the rapid sea-level rise associated with Meltwater Pulse 1A (Weber et al.,







2014). This transgression submerged vast areas of the Patagonian continental shelf, significantly reducing the availability of exposed, deflatable sediments and thereby curtailing PAT dust emissions to Antarctica. Simultaneously, the progressive deglaciation of the Patagonian Ice Sheet triggered substantial changes in regional riverine systems. New drainage pathways opened up effectively redirecting large portion of river catchment from the Atlantic to the Pacific Ocean, hence decreasing sediment supply and changing the planform of rivers flowing on the eastern flank of Patagonia (Thorndycraft et al., 2019; Davies et al., 2020; Skirrow et al., 2021; Gaiero et al., 2003; Caldenius, 1932). These hydrological and geomorphological transformations further limited the potential for dust production in Patagonia. The combined effects of continental shelf submergence in Patagonia (and New Zealand), along with drainage reversals and reorganized river networks, led to a persistent decline in high-latitude dust contributions. Those phenomena induced a relative increase in dust input from lowerlatitude source regions such as Australia, Southern Africa, and the Puna-Altiplano, further enhanced by the long-term trend of increasing aridity in these regions and the greater efficiency of long-range atmospheric transport from low-latitude to Antarctica (see above). Importantly, our data demonstrate that these feedback mechanisms influenced dust deposition not only in EDML (in the Atlantic sector of East Antarctica) but also in EDC, thus extending their impact across to the Indian Ocean sector. This highlights the spatially extensive and long-lasting effects of eustatic sea-level rise and postglacial hydrological reorganization on the composition of Antarctic dust assemblages during the LGIT.

## 5. Conclusion

This study presents the first high-resolution, continuous and quantitative reconstruction of dust provenance in the EDC ice core over the Last Glacial-Interglacial Transition (LGIT). Our results reveal a distinct shift in dust source composition from glacial to interglacial periods. During the MIS3, LGM and Heinrich Stadial 1, dust reaching EDC was predominantly derived from high-latitude glaciogenic sources, particularly Patagonia (and New Zealand), reflecting the extensive outwash plains, cold and arid conditions, and efficient dust mobilization mechanisms in those regions. From ~14.5 kyr BP onward, a marked reorganization in dust provenance occurred, characterized by a relative decline in high-latitude PSA contributions and a corresponding rise in dust from low-latitude sources such as Australia, the Puna-Altiplano Plateau, and Southern Africa. As such those results confirm our previous study on the EDML dust record showing similar shift that we ascribed to a complex interplay of hydrological rearrangement processes in Patagonia and the rapid submersion of its shelf. This shift in provenance is effectively decoupled from the massive decline in dust deposition that occurred between 18 and 15 kyr BP due to changing atmospheric transport dynamics. Comparisons with the EDML record reveal both broad similarities and specific key regional differences in dust provenance across East Antarctica: while millennial-scale changes were broadly synchronous, the relative contributions of PAT, SAF, NZ and AUS differed reflecting regional climatic dynamics in their respective source areas. Finally, our findings emphasize the role of glacial-eustatic and hydrological feedbacks—particularly

sea-level rise, drainage reversals, and paleolake desiccation—in reshaping dust source activation and transport. These processes, in turn, contributed to the persistent reorganization of the Southern Hemisphere dust cycle during the LGIT, with implications for biogeochemical fluxes, atmospheric circulation (chemistry and physics), and climate feedbacks on glacial-interglacial timescales.

CODE/DATA AVAILABILITY. All data needed to evaluate the conclusions in the paper are present in the paper and/or the Supplementary Materials. Additional data related to the paper may be requested from the authors. Correspondence and requests should be addressed to Si.B.

**AUTHOR CONTRIBUTION.** P.B performed the measurements; A.V., St. B and G.L. developed together the model; Si.B and St.B. analyzed the data and wrote the manuscript draft; All authors reviewed the manuscript.

**COMPETING INTERESTS.** The authors declare no competing interests.

APPENDIX S. SUPPLENTARY DATA. Supplementary data to this article

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

Figures

Figure 1: Evolution of dust PSA contributions to EDC ice core between 2852 to 33699 yr BP (% of total dust deposition for each PSA – Table S1). Bold lines denote the mobile average over 9 steps. Central-Western-Argentina and Antarctica are not shown as their dust emissions contribution are very low. Gray vertical zones delineate the various periods of the transition from end of MIS3 to Holocene.

Figure 2: Compilation of <sup>87</sup>Sr/<sup>86</sup>Sr and εNd isotopic compositions representative of PSA, dust in ice cores (refs. Table S1, - open symbols) and calculated isotopic values using REE contributions from EDC– solid dots (see section 4.1 for details). Colored-frame areas denote the domains of isotopic values of respective PSA used for the isotope calculation i.e., involved in the dust assemblage (data from Grousset et al., 1992; Delmonte et al., 2004; Gingele and De Deckker, 2005; De Deckker et al., 2014; Revel-Rolland et al., 2006; Gaiero, 2007; Gili et al., 2017; Gili et al., 2022; Li et al., 2020; Koffman et al., 2021). Note that the SAF and AUS domain is more extended than showed in the Figure. REE-based calculated isotopic composition (blue to red dots corresponding to periods of LGIT) were obtained from (i) the % contributions of PSA and (ii) the isotopic fingerprint (Table S4) and elemental concentration of the PSAs end-members (Table S3).

Figure 3: Box plots of provenance contributions (in %) of Holocene, and Last Glacial Maximum (LGM) dust depositions in EDC ice core. Horizontal lines in the boxes indicate the median values of contribution for each source, cross symbols denote the mean and the extremes of the colored boxes indicate the first and third quartiles, and the whiskers indicate the lower and upper extremes. Box plots for HS1, ACR and YD are illustrated in Figure S4.

Figure 4: Histograms representing the contribution percentage difference (EDC - EDML) averaged over periods for each PSA between EDC and EDML ice cores (Table S4).

Figure 5: Comparison of the contribution of Patagonian dust (PAT) to EDC ice core (A) and (B) the Median Destructive Field of isothermal remanent magnetization (MDF) of lacustrine sediments in Laguna Potrok Aike (Lisé-Pronovost et al., 2015). Bold lines denote the mobile average over 9 steps.