# Peer review of "Quantifying Southern Hemisphere dust sources during the Last Glacial-Interglacial Transition using rare earth elements in the EPICA Dome C ice core"

_EGUsphere, 2025_

## Referee Comment (RC1)

**Boxho et al. "Quantifying Southern Hemisphere dust sources during the Last Glacial-Interglacial Transition using rare earth elements in the EPICA Dome C ice core"**

**Reviewed by Austin Carter**

This paper presents a high-resolution record of source contributions to embedded dust within the EPICA Dome C (EDC) ice core, specifically focused across the time interval of 33.7-2.9 kyr BP. The authors apply a statistical model previously developed by Vanderstraeten et al., 2023 to a large dataset of discrete rare earth elemental (REE) concentrations of ice core dust from EDC as reported in Gabrielli et al., 2010. The previously published dataset contains nearly 300 sections of ice from EDC which have been acid leached using 1% $HNO_3$ and individually characterized for REE concentrations. Subsequently, the model in Vanderstraeten et al., 2023 includes a database of REE concentrations for a variety of samples from potential sources within the Southern Hemisphere. After making the EDC leached REE data comparable to total REE concentration using element-specific enrichment factors, the authors then modeled various combinations of sources that would result in the most closely matched data for each ice core dust sample. In this way, source contributions are effectively "unmixed" from the dust.

The authors show that before 14.5 kyr BP high-latitude PSAs (Patagonia + New Zealand) dominate the EDC ice core dust record and are stable. After 14.5 kyr BP, there is more heterogeneity in PSA contributions with periods of greater low-latitude PSA contributions (Australia, South Africa, and Puna-Altiplano Plateau). These changes are attributed to both hydrological changes in Patagonia and its corresponding shelf exposure as well as regional source dynamics at low-latitude sources. Candidly, I found the latter to be a bit hard to follow in its current organization. Lastly, when compared with EDML, the EDC record's PSA contributions over time are broadly similar with subtle differences owing to their geographic positioning. Overall, I enjoyed reading and learning more about this statistical application, and I find it very clever and interesting; however, I have some comments on clarity and the inherent limitations in the methods that I believe merit additional discussion in the paper after which I believe it would have my full support. I hope the following suggestions can help improve the manuscript:

**Specific comments**

L17: The authors frequently refer to their record as "continuous." However, it is my understanding that the EDC REE dataset from Gabrielli et al., 2010 used for this study is not continuous. Instead, these measurements were on discrete sections of ice. If this is correct, then I do not think it's valid to describe the paper's results as continuous, and this should be edited for clarity.

L34-52: The opening paragraph of the introduction parallels that of the opening paragraph of Vanderstraeten et al., 2023 very closely with regard to sentence structure and reference order etc. While I recognize the parallel logic between the papers, it would be nice to differentiate this manuscript's introductory paragraph a bit further.

L69: It would be useful to elaborate or define what is meant by "REE patterns" here.

L121-124: The first limitation when applying the DEPOT statistical model lies in the ability to compare acid-leached REE concentration data with other REE concentration data from PSAs. There is a possibility for preferential leaching of LREE (La, Ce, Pr, Nd) during a 1% HNO3 leach as shown in Gabrielli et al., 2010. Hence the need to apply enrichment factors to translate data from acid-leached to its potential total concentration form, which I think is handled well in the manuscript. However, the readers are left to assume that all of the PSA data are in total REE concentration form. Can you confirm this? Or are enrichment factors also applied to these data in their respective studies? I think some added clarity and its importance for direct comparisons would be valuable and would strengthen the paper's methods.

L129: While I recognize the term Dust Sources (DS) was previously adopted in Vanderstraeten et al., 2023, I do think it can be slightly misleading. From my understanding, these are not unique sources, rather these are individual samples of dust, loess, sediment, and/or soil that make up a Potential Source Area (PSA). The text would benefit from some clarity on that and additional descriptions here. The confusion is made clear when we take Patagonia, for example. The authors describe 39 Dust Sources that make up the Patagonia PSA. To me this implies, 39 unique regions, but I only count 19 markers on Fig. S2 for Patagonia. This is likely because multiple samples come from similar sites, which therefore would not necessarily be a different dust source.

L127-133: The second and most important limitation in this approach is related to the quantity of data in the PSA database. Because the goal is to compare all samples in the PSA database individually to the ice core samples, PSAs that have a greater quantity of REE concentrations might have a bias. Perhaps this is less of an issue after taking the median of each PSA-specific PDF, but it is hard to know without seeing an example. I made a pie chart showing the distribution of the PSAs below. Australia makes up nearly half of the database, which I'm concerned that the % contributions might be biased heavily as a result. Additionally, based on Pb isotopic data for both EDC ice core dust and PSAs, Gili et al., 2016 argues against significant glacial and interglacial Australian contributions to the EDC dust record, yet this paper's finding suggests otherwise. If there is no bias in the DEPOT model, then the findings showing large influences from Australia are interesting and worth discussing further in that regard. While I agree with L366-355, importantly this study's findings with respect to Australia are not fully consistent with those in Gili et al., 2016.

Similarly, I believe the Antarctic PSA might be discounted too easily and quickly despite it being argued as a plausible contributor during the Holocene at EDC in Gabrielli et al. 2010 from which the REE dataset originates. Additionally, Delmonte et al., 2013 highlights how the measured EDC ice core dust for the Holocene are geochemically similar in Sr-Nd isotope composition to Antarctic PSAs. However, the PSA database contains only 5 samples from the McMurdo Dry Valleys to represent the Antarctic continent (though only 4 are reported in Table S2). Because there are so few available samples, it seems inherently less likely that there will be enough trends to compare to and, thus, less likely they will make up a significant portion of the EDC ice core dust in the statistical model. Part of this issue may also be a result of limited measured Antarctic PSAs for REE concentrations within the literature. Some added caution here would be valuable and perhaps the inclusion of dust size might be useful as well.

Along the same lines, Coppo et al., 2022 argues that the activation of S-CWA is a plausible dominant PSA to the East Antarctic Plateau during the LGM; however, this paper's

findings suggest it is not a dominant contributor during any time across the LGIT. This is an interesting difference and may also be related to its limited quantity (n=5) in the PSA database. All of these above points, especially when in contrast to papers that are heavily cited in the text, are worth noting/discussing further.

[Figure]

L258-260: I agree with the principle of modeling Sr-Nd isotope compositions and comparing it to established data as a fidelity check for the DEPOT algorithm and thank the authors for exploring this; however, I have some concerns in the way this is currently presented in the manuscript.

- First, the measured Sr-Nd isotope compositions of samples are described as coming from dust in ice cores as reported in previous studies; however, the Iso Early Holocene category appears to consist predominantly of dust from modern snow pits on Berkner Island (not from ice cores; Bory et al., 2010). Furthermore, in the text Early Holocene is classified as 11.7-7.5 kyr BP, but these Berkner Island data do not represent that time interval. The dust described in Bory et al., 2010 span a two-year period from 2002-2003. Therefore, I do not think these are relevant and should be removed.
- Second, after removing the Berkner Island data, I would not necessarily agree that there is close agreement between the modeled EDC Sr-Nd isotope compositions and the literature measured Sr-Nd isotope compositions of dust from Dome C and Taylor Glacier. For East Antarctic sites during the Holocene, $\varepsilon_{Nd}$ values of ice core dust are generally below 0 but not lower than -10 and for $^{87}Sr/^{86}Sr$ range approximately between 0.707 and 0.715. While the modeled Holocene data do capture a wider range than the modeled LGM data (as expected), the modeled Holocene data appears to extend beyond what would be expected from measured ice core dust, which merits some explanation. Similarly, the modeled LGM measurements span a wider range than would be expected for MIS 2 samples, which based on the literature should be tightly clustered around $\varepsilon_{Nd}$ of -1 or -2 and $^{87}Sr/^{86}Sr$ of ~709. Furthermore, when looking at the PSA % contributions during the LGM, all PSAs appear relatively stable and with a consistent mix of sources, I

would expect a much tighter modeled cluster as a result. Why might be the modeled data result in these wider ranges?

- Lastly, I think part of the difficulty in deciphering Figure 2 is that there are multiple goals rooted in several comparisons (i.e., comparing time interval changes, modeled vs. measured data, and PSAs). Perhaps this could be split into multiple figures or panels. One could focus on directly comparing modeled vs. measured data and one on just comparing modeled data to PSAs.

Section 4.3: This section is packed with useful information but does struggle a bit in its structure. The opening paragraph provides three processes to explain the variability across the Holocene. As a reader, I expected the subsequent paragraphs to be focused on a discussion of these processes in greater detail, but that was not necessarily the case. The next paragraph (L315-334) feels very results heavy and compares the record broadly to other papers. Then, the following paragraphs are subdivided into PSA-specific discussions, which are very detailed but also hyper-focused. I think this section would be greatly improved by a slight restructuring and including a broader discussion of the glacial-eustatic and hydrological feedbacks that connect the regional source dynamics.

L467: I don't think the data necessarily "demonstrate" unequivocally that these feedback mechanisms influenced dust deposition rather they "support" that scenario or "suggest" it, which may be better phrasing.

L480-482: The "hydrological rearrangement processes in Patagonia and the rapid submersion of its shelf" would explain the decline in high-latitude PSA contributions but probably not the rise in dust from low-latitude sources? I think the current phrasing implies the latter is providing confirmation when it may not actually be related.

**Figure modifications**
- Figure 1
  - It would be best to change the units on the x-axis to kyr to be consistent with descriptions in the text and the same for the caption
- Figure 2
  - The marker colors appear identical to the colors used for the different domains. Because of this, at first glance, it seems like the markers would represent PSA data rather than ice core dust. I would suggest exploring additional color schemes or styles to make this a bit clearer.
  - It is not obvious from looking at the figure legend which samples are measured data vs. modeled data. To add clarity, one suggestion would be to include legend titles for each column. For example, one might say "EDC Modeled (this study)" and "Measured (previous work)." It also should be made clear in the figure caption which sites the measured data come from. Notably, there are only 3 samples shown which are measured from the Dome C area; the majority of which come from the coastal and low-elevation Taylor Glacier and snow pits on Berkner Island.

- If you are specifically looking for ice core dust only, below are some paper suggestions in which you may find additional data relevant to the study's time interval:
  - Aarons et al., 2016: Taylor Dome (MIS 2-1)
  - Delmonte et al., 2010a and 2010b: Talos Dome (MIS 3)
  - Delmonte et al., 2010b: Talos Dome (MIS 2-1)
  - Delmonte et al., 2007: Dome C and Vostok (MIS 1)
  - Delmonte et al., 2004b: Dome B and Komsomolskaia (MIS 2)
  - Delmonte et al., 2017: Dome B and Old Dome C (MIS 2)
- Technical comments
  - The compilation is listed as in "Table S1" but this should be "Table S4."
  - Edit: "Note that the SAF and AUS domains are more extended than shown in the figure."
  - $^{87}Sr/^{86}Sr$ ratios are shown with commas instead of periods, but in the text are referred to using only periods. This should be changed to be consistent with the text.

- Figure 5
  - The x-label is missing.
  - Panel B could benefit from some additional discussion in the text just to elaborate further on why its inclusion is relevant.
- Figure S3
  - Missing units on the y-axis
  - Would be helpful to make the x-axis kyr to be consistent with the text

**Technical corrections**
Epsilon notation should be briefly defined somewhere in the text and should be displayed consistently. Of note, it appears three different ways throughout the text and figures: $\varepsilon_{Nd}$, $\varepsilon Nd$, and $\varepsilon(Nd)$.

The age of events and ranges should be shown in a consistent format both in units and in chronology. For example, L109-110 provides a range spanning young to old and in yr BP but previously this has been described spanning old to young and in kyr BP.

The use of hyphens, en dashes, and em dashes are used interchangeably often and in the same paragraphs. For example, L293 and L301 as well as L185 and L188. I would suggest combing through and editing this.

Sometimes percentages are written X% and sometimes as X % with an additional space. I would suggest removing this extra space from all percentages.

Typing errors
- L62: is → are
- L77: reveal → revealed
- L101: camples → samples
- L146: step. The → step, the
- L147: la-lu → La-Lu

- L193: averages → average
- L199: low → light
- L220: Epica → EPICA
- L236: describes → described
- L246: e.i. → i.e.
- L249: represents → represented
- L261: persistance → persistence
- L374: kry → kyr
- L458: planform → platform?
- L501: supplentary → supplementary
- Table S1 caption: th → the

Author Contributions: I'm assuming P.B. is supposed to be P.G.? Also, I think this section is missing N.M. and A.B. contributions.

Throughout the references, there are some issues with what appears to be accented characters. For example, see Markgraf et al., 2007.

---

## Referee Comment (RC2)

Review of Boxho et al, CP
Dec. 30, 2025 by Bess Koffman

This paper applies a previously developed statistical 'unmixing' model to a previously published rare earth element (REE) dataset from the EPICA Dome C ice core in Antarctica to infer changing dust source contributions over time. In general, I think this is a worthwhile exercise with the potential to clarify broad interpretations about changing dust provenance during climate transitions, and their causes. Following major revisions, it should be appropriate for publication in *Climate of the Past*.

The paper needs substantial work to be publishable. My main critiques are that 1) the estimates of potential source area (PSA) contributions are presented without uncertainties, making them basically meaningless; 2) differences in interpretation between this study and previous ones are not thoroughly described and supported with evidence; 3) the Younger Dryas is included in the interpretation as if it were a climate event in Antarctica; I find this baffling given the geographical focus of the paper; 4) figures have a range of issues (see comments below); 5) the writing needs major revision with respect to grammar, subject/verb agreement, punctuation, and spelling. In addition, the introductory paragraph follows the structure and citation ordering of the Vanderstraeten et al 2023 study from the same group (thanks to the other reviewer for pointing this out). While I wouldn't call it plagiarism, the similarity is striking. This paragraph should be rewritten and additional citations included.

Regarding point 1) above: in the paragraph starting on Line 153, analytical uncertainties are discussed, along with their impact on the model outputs, but the uncertainties on the dust provenance estimates from the MC simulations themselves are not given anywhere (unless I missed them somehow). This seems like a simple correction but a very important one. Uncertainties should be incorporated into the text where numerical values are given, as well as into the figures where temporal changes in PSA contributions are indicated (i.e. as colored confidence intervals).

Regarding point 2) above: there are a number of places where comparisons to prior publications (and their interpretations) are given in a cursory manner, but differences are not fully explored or justified. Considering the volume of published papers on Antarctic dust provenance, this is a place where the present study could really aim to leverage its findings for new insights.

One example relates to South American dust sources. The explanation of why the southern Puna and southern Altiplano both contribute dust but not the northern Puna (which is geographically sandwiched between them) needs to be clarified and better justified. This is especially confusing because in Fig. 3a of Gili et al 2017, the REE fields of the northern and southern Puna fully overlap, and their paper includes N Puna as a source. What specific REE ratios support eliminating N Puna as a dust source to EDC? I would also like the authors to reference and/or address the statement in Gili et al 2017 that "REE are less useful for distinguishing sediments from CWA and Puna." The Gili paper highlighted CWA as a more important source than Patagonia, which I don't really see addressed in the present study. Also, a number of studies have treated Tierra del Fuego as its own source area, but the present study seems to ignore it. Is it simply not included, or is TdF being incorporated into the Patagonia PSA? Some discussion is

warranted – and ideally, the paper would provide full treatment of TdF as its own source region. A more in-depth discussion of the differences between this study's conclusions and those of Gili et al 2017 is needed, with evidence provided to support the interpretations.

Minor comments:

Line 31: "Linking dust composition to eustatic sea level rise" – This isn't new as other studies have highlighted this potential link before. Also, I don't see any evidence presented in this paper that conclusively links dust provenance changes to any specific process in South America such as those described here. Either these connections need to be strengthened, or this sentence should be cut from the abstract.

Line 40: this statement deserves a more complete set of citations.

Line 88: Remove hyphens in this sentence.

Line 123: Please provide more information on the digested samples: how many were there? At what temporal resolution? Are they evenly distributed throughout the part of the core used for this analysis?

Line 127: Where are the citations for these data provided? Are the published PSA data all on digested samples? Needs to be fully described here, as differences in leaching/digestion could lead to discrepancies in the resulting REE concentrations.

Line 130: Why is Tierra del Fuego not included?

Lines 145-146: needs revision.

Line 179: Please omit the dash in this sentence.

Line 187: the time period listed as "Late Holocene" is really more "Middle Holocene," especially considering the cutoff at 2.8 ka. Further, the "Early" and "Middle" portions of the Holocene are commonly distinguished at 8.2 ka. The authors might want to consider this date as the boundary rather than the seemingly arbitrary 7.5 ka.

Line 189: remove the word "for"

Line 199: What does "mafic-like" mean? Can you just use the word "mafic"? Also, LREE stands for "light rare earth elements" not "low". Please correct this.

Paragraph beginning line 201 and throughout: What are the uncertainties on these estimates of PSA contributions? Please give the plus-or-minus values. It's hard to assess how meaningful these numbers are without quantified uncertainties.

Line 220: Please capitalize EPICA.

Lines 202 and 223: The listed PSAs have a different order in these two sentences but it is not obvious that they refer to different time periods. Please double-check what is correct and revise as needed.

Line 258: The authors state there is "close agreement" between the measured and modeled Sr-Nd isotope values, but Figure 2 suggests the agreement during certain intervals is better than others. The modeled "iso" symbols appear to be systematically offset to lower 87Sr/86Sr from the measured values in the eNd range of ~ -7 to -12. Then in the LGM interval, the "iso" symbols are tightly clustered in a narrow Sr-Nd range that is also offset to higher 87Sr/86Sr values compared to the bulk of the actual data. The authors need to address the reasons for these discrepancies between the REE-inferred Sr-Nd isotope data and the measured values.

Line 261: "Persistence" is misspelled.

Line 283: It might be helpful to the reader to provide some context, e.g. by adding "on the basis of combined Sr-Nd-Pb isotope compositions" to the end of this sentence.

Line 284-285: Koffman et al 2021 also estimated the expanded outwash plain area available for dust deflation during the LGM with a sea level lowering of 130 m. Their estimate, at ~75,000 $km^2$ of exposed continental shelf, is a bit higher than that of Vanderstraeten et al 2023. It might be worth providing both values to give some sense of the range of estimates.

Line 306: The YD is mentioned here without context. Why is this Northern Hemisphere climate phenomenon relevant to a paper on Antarctic dust provenance? If the paper were focused on interhemispheric climate signals and phasing I could see the logic for highlighting the YD, but otherwise it just seems out of place. I suggest focusing on Southern Hemisphere climate signals such as the ACR in this paper, as this seems more relevant.

Line 307: The sentence beginning on this line needs grammatical revision.

Line 311: The double angled brackets around "wet scavenging" can be removed.

Line 334: The year for the Speirs article is 2010, not 2001. Further, the direction the winds blow in the McMurdo Dry Valleys, as shown in that article, is generally west-to-east, not east-to-west (see e.g. their figure 4, showing winds from 270 degrees). One must remember that north is "up" toward the coast when looking at maps of Antarctica. Please correct this sentence.

Lines 339-352: It is not possible to evaluate this paragraph given the lack of labeling on Fig. 5

Line 353-362: The explanation of why the southern Puna and southern Altiplano both contribute dust but not the northern Puna (which is geographically sandwiched between them) needs to be clarified and better justified. This is especially confusing because in Fig. 3a of Gili et al 2017, the REE fields of the northern and southern Puna fully overlap, and their paper includes N Puna as a source. What specific REE ratios support eliminating N Puna as a dust source to EDC? I would also like the authors to reference and/or address the statement in Gili et al 2017 that "REE are less useful for distinguishing sediments from CWA and Puna." Also, a number of studies have

treated Tierra del Fuego as its own source area, but the present study seems to ignore it. Is it simply not included, or is TdF being incorporated into the Patagonia PSA? Some discussion is warranted – and ideally, the paper would provide full treatment of TdF as its own source region. And finally, a more in-depth discussion of the differences between this study's conclusions and those of Gili et al 2017 is needed, with evidence provided to support the interpretations.

Line 384-385: A phrase is repeated

Line 407: Why are the ACR and YD included together as if they are both climate events in Antarctic ice cores? I suggest removing YD references unless there is clear justification for discussing the YD.

Line 426: remove second use of "dust" in this sentence

Line 430: "PSA sources" is redundant. Can simply use "PSAs". This sentence also should include citations as this is very detailed information about transport pathways.

Line 466-470: This is very arm-wavy given the actual evidence presented in this paper. Suggest removing or toning down the language here.

Line 476: The contributions of Patagonia and New Zealand should be given summarily here. If NZ is considered a major source, then it should not be in parentheses.

Line 488: More evidence would need to be presented to support this statement. I do not believe the paper as written provides clear evidence of any of these processes or feedbacks beyond loose temporal correlation. I would like to see a more robust treatment of the range of potential mechanisms (including robust citations) driving changes in dust delivery to EAIS in order to support a statement such as this.

Line 497: I believe "P.B." should be "P.G.," Paolo Gabrielli.

Figures

Fig. 1: Please change the x-axis time units to years or thousands of years. Showing time in units of 10,000 years is atypical and not intuitive. This figure would be greatly improved by showing the dust flux in the ice core in addition to the individual PSA contributions. It would also be very helpful to see a d18O or dD record for climate context, particularly to emphasize temperature variations during the deglaciation and to compare to changes in dust deposition and provenance. I also think the "YD" highlighting should be removed, as it does not seem relevant to the study.

Fig. 2: In general, I question the "blobs" as currently drawn. The Patagonia field appears far too wide given the data published from this region. The Australia field is missing data from the Northwest Territory and South Australia that would deepen the field to much lower eNd values than what is shown here (e.g. De Deckker 2019). The Southern Puna region has a highly improbable field as drawn; it should be more convex around the available data. The extremely high $^{87}Sr/^{86}Sr$ value included in the NZ field is likely erroneous. I strongly suggest including only

data with well-characterized geologic and geomorphic context, in this case, Koffman et al. 2021. In general, I would like to see the actual data points used to generate these "blobs" and the specific citations included for each. This could be a supplementary figure that supports the main text, for instance. But I also think revision of this figure is warranted.

Fig. 5: I understand the motivation for this type of comparison, but if the current study aims to draw broader interpretations about the westerlies, or to use proxy records of the westerlies to help interpret the presented dust provenance record, there needs to be appropriate context and breadth of records included. If a figure of this type is to be included, I would also like to see other records from South America and from Antarctic ice cores for context here, as I think it will really strengthen the interpretations and enhance the impact of this work. For instance, you might want to include the NPI (*Nothofagus* to *Poaceae* Index) from Lago Guanaco (Moreno et al. 2010, the Macquarie Island diatom-inferred conductivity record of Saunders et al. 2018, the opal upwelling proxy record of Anderson et al. 2009, the CO2 data and dD data from EPICA Dome C, etc.

In addition, it seems to me that dust flux would be more meaningful to compare to Potrok Aike and these other records. Percent contribution, as currently shown, is a factor of the relative inputs of multiple sources and climate and environmental conditions in those regions, so is less meaningful. It would be better, I think, to scale the total dust flux by the percent from Patagonia and use that in this figure instead (e.g. Patagonia dust flux to EDC). Please also correct the x-axis to be in years or ka, and to match the timescale shown in Fig. 1. Here the (x10$^4$) is missing so I think the plot is completely uninterpretable as shown. It also needs an x-axis label with units stated. The colored bars are not labeled or described in the caption, but need to be.

Fig. S2: Please fix the x-axis as in other figures and include appropriate units and labels.

Fig. S5: Large y-axis label contains a typo. Actually, this figure is pretty interesting and might be worth adding to the main text once it is revised. I think it provides a nice complement to Fig. 4. I note that the x-axis ages and labeling are distinctly different from the other figures. As mentioned before, figures should all use the same units – either years or kyr.

Table S4 and elsewhere: "Localisation" in English means to make something more localized. It would be better to use the word "Location."

All Tables and Figures: Please use decimals for numbers rather than commas.

References cited:

Anderson, R. F., Ali, S., Bradtmiller, L. I., Nielsen, S. H. H., Fleisher, M. Q., Anderson, B. E., & Burckle, L. H. (2009). Wind-driven upwelling in the Southern Ocean and the deglacial rise in atmospheric CO2. *Science, 323*, 1443-1448. doi:10.1126/science.1167441
De Deckker, P. (2019). An evaluation of Australia as a major source of dust. *Earth-Science Reviews, 194*, 536-567.

Gili, S., Gaiero, D. M., Goldstein, S. L., Chemale Jr, F., Jweda, J., Kaplan, M. R., . . . Koester, E. (2017). Glacial/interglacial changes of Southern Hemisphere wind circulation from the geochemistry of South American dust. *Earth and Planetary Science Letters, 469*, 98-109.

Koffman, B. G., Goldstein, S. L., Winckler, G., Borunda, A., Kaplan, M. R., Bolge, L., . . . Vallelonga, P. T. (2021). New Zealand as a source of mineral dust to the atmosphere and ocean. *Quaternary Science Reviews, 251*. doi:10.1016/j.quascirev.2020.106659

Moreno, P. I., Francois, J. P., Moy, C. M., & Villa-Martinez, R. (2010). Covariability of the Southern Westerlies and atmospheric CO2 during the Holocene. *Geology, 38*(8), 727-730.

Saunders, K. M., Roberts, S. J., Perren, B., Butz, C., Sime, L., Davies, S., . . . Hodgson, D. A. (2018). Holocene dynamics of the Southern Hemisphere westerly winds and possible links to CO2 outgassing. *Nature Geoscience, 11*(9), 650-655. doi:10.1038/s41561-018-0186-5

Vanderstraeten, A., Mattielli, N., Laruelle, G. G., Gili, S., Bory, A., Gabrielli, P., . . . Bonneville, S. (2023). Identifying the provenance and quantifying the contribution of dust sources in EPICA Dronning Maud Land ice core (Antarctica) over the last deglaciation (7–27 kyr BP): A high-resolution, quantitative record from a new Rare Earth Element mixing model. *Science of the Total Environment, 881*, 163450. doi:https://doi.org/10.1016/j.scitotenv.2023.163450